Spammer group detection and diversification of customers’ reviews

http://orcid.org/0000-0003-3874-2088 Hussain Naveed 1 2
Mirza Hamid Turab 1 drturab@cuilahore.edu.pk
Ali Abid 1
http://orcid.org/0000-0002-5536-8764 Iqbal Faiza 2
http://orcid.org/0000-0001-6118-0314 Hussain Ibrar 2
Kaleem Mohammad 3
1 Department of Computer Science, COMSATS University Islamabad, Lahore Campus , Lahore , Pakistan
2 Department of Software Engineering, The University of Lahore , Lahore , Pakistan
3 Department of Electrical Engineering, COMSATS University Islamabad , Islamabad , Pakistan
Alatas Bilal
Electronic publication date: 2021 Apr 9
Publication date: 2021
Volume: 7
Electronic Location ID: e472
Received 2020 Nov 4; Accepted 2021 Mar 15
Copyright: © 2021 Hussain et al.
Copyright year: 2021
Copyright holder: Hussain et al.
License: This is an open access article distributed under the terms of the Creative Commons Attribution License, which permits unrestricted use, distribution, reproduction and adaptation in any medium and for any purpose provided that it is properly attributed. For attribution, the original author(s), title, publication source (PeerJ Computer Science) and either DOI or URL of the article must be cited.
License URL: https://creativecommons.org/licenses/by/4.0/

Keywords: Online customer reviews, Products and services reviews, Spam review detection, Spammer group detection, Spammer behavioral features, Review diversification

Funding: The authors received no funding for this work.

==============================
Online reviews regarding different products or services have become the main source to determine public opinions. Consequently, manufacturers and sellers are extremely concerned with customer reviews as these have a direct impact on their businesses. Unfortunately, to gain profit or fame, spam reviews are written to promote or demote targeted products or services. This practice is known as review spamming. In recent years, Spam Review Detection problem (SRD) has gained much attention from researchers, but still there is a need to identify review spammers who often work collaboratively to promote or demote targeted products. It can severely harm the review system. This work presents the Spammer Group Detection (SGD) method which identifies suspicious spammer groups based on the similarity of all reviewer’s activities considering their review time and review ratings. After removing these identified spammer groups and spam reviews, the resulting non-spam reviews are displayed using diversification technique. For the diversification, this study proposed Diversified Set of Reviews (DSR) method which selects diversified set of top-k reviews having positive, negative, and neutral reviews/feedback covering all possible product features. Experimental evaluations are conducted on Roman Urdu and English real-world review datasets. The results show that the proposed methods outperformed the existing approaches when compared in terms of accuracy.

Introduction

Customer reviews have become the major source to collect different opinions about products and services. These reviews can affect daily life decisions and professional activities: e.g., which restaurant is good, which car to purchase, which product to buy and which doctor to consult. Online reviews can be posted as genuine reviews or spam reviews. Spam reviews are usually written by individuals/spammers to highlight any product or service by giving spam reviews. Spam reviews may directly interpret financial advantages and losses for a company. For example, the large number of favorable reviews about products and services may attract more customers and negative reviews are often a reason for decline in the sale (Tang, Qian & You, 2020). Distinguishing fake reviewers from the genuine reviewer in an online forum is a challenging and open research issue. Therefore, in recent years, the Spam Review Detection (SRD) problem has gained much attention in the research community. It is believed that without solving this significant problem, the review websites could become a place full of lies and completely useless (Gong et al., 2020). A recent survey suggests that online reviews for purchasing products or services play a critical role in the decisions of potential buyers (Ren & Ji, 2019). It has been observed that 70 percent of customers trust in the reviews written by other customers, while 90 percent read these reviews before making financial decisions (Mintel, 2015; Luca, 2011).

A spammer group refers to a group of reviewers who works together for writing spam reviews to promote or demote a set of targeted products. Spammer groups are very damaging as they can produce a huge amount of spam reviews collectively. When a group is working collaboratively towards a product, it can take control of the sentiment of the customer for the product. The group of reviewers is usually represented as a set of reviewer-IDs. These IDs can belong to a single person having multiple IDs, multiple persons with single IDs and/or combination of both (Vidanagama, Silva & Karunananda, 2020).

In the last few years, writing spam reviews has become a business. Spammers do not write a single review. They usually write many spam reviews for different products to earn money. Therefore, collective behaviors of a group working together on several products can be difficult to identify the spam reviews. Most of the existing studies have identified spam reviews by utilizing the individual spammer behavioral features (Cao et al., 2019; Hussain et al., 2020; Rahman et al., 2015).

To the best of our knowledge, the spammer group detection task has not been thoroughly studied for the South Asian languages, specifically for Roman Urdu reviews. Roman Urdu is used for the Urdu language written in Roman script (using English alphabets) (Fatima et al., 2017). For example, Urdu sentence:

“موبائل کی بیٹری کا دورانیہ بہت کم ہے اور اس لحاظ سے اس کی قیمت زیادہ ہے”

will be written in Roman Urdu as “Mobile ki battery ka dorania both kam hai aur is lehaz se es ki qeemat both ziada hai” whereas in English it will be translated as “Mobile’s battery duration is too short, and compared to this its price is very high.” Roman Urdu uses English alphabets and Urdu semantics. The first objective to conduct this research is to identify spammer groups and spam reviews in Roman Urdu reviews and English reviews to increase customer confidence regarding product purchases in the South Asia, particularly in Pakistan.

It is a common practice that the reviewers usually read two or three reviews from the top reviews and decide about the purchase of the product/service. These top reviews usually contain positive/negative reviews and may not present a complete view of all reviewer’s feedback (Zhang et al., 2020). Figure 1 presents an example scenario of reviews and compares the presentation of non-diversified reviews with diversified reviews. Figure 1A displays all the positive top-k reviews regarding a mobile model which shows the bias toward promoting the product whereas Fig. 1B presents an assorted representation of positive, negative, and neutral reviews covering all possible features of the same mobile model. The diversified reviews are more reliable and useful for the user to make the decision (Abid et al., 2016). The second objective of this research is to present a diversified view of top-k non-spam reviews having positive, negative, and neutral sentiments covering all possible product features.

Figure 1 Example of reviews.

(A) Positive reviews. (B) Diversified reviews.

This study proposed a novel Spammer Group Detection (SGD) method to find out suspicious spammer groups that work together to produce spam reviews. SGD works in several phases. First, it produces co-reviewer graphs for identification of reviewers having similarity in reviews based on review post time and review rating. Then, it utilizes the Structural Clustering Algorithm for Networks (SCAN) algorithm to find out candidate spam groups. Next, it analyzes various individual spammer behavioral features and group spammer behavioral features to calculate the spam score of each candidate group reviews. Finally, using a predefined threshold value, the group having a higher spam score than the threshold is labeled as spam group and assumed that all reviews written by that group are spam. The training and testing of the proposed SGD method are conducted using deep learning classifiers: Convolutional Neural Network (CNN), Long Short-Term Memory (LSTM), Gated Recurrent Units (GRU), and Bidirectional Encoder Representations from Transformers (BERT).

In addition to finding non-spam reviews using SGD method, this work also presents a novel diversification method; named as Diversified Set of Reviews (DSR), which selects diversified set of top-k non-spam reviews having positive, negative, and neutral reviews/feedback covering all possible product features. DSR method is used to present non-spam reviews which are identified by SGD method. DSR approach works in several phases, first, review dataset of the product or service is divided into three categories based on positive, negative and neutral sentiments, then, the diversified feature extraction process is applied for each product or service such that expanded features are extracted. In the end, all three diversified categories of datasets are combined to display top-k diversified non-spam reviews of particular product or service having varied sentiments and expanded product features. The DSR method save time for the reviewers to decide about product and services without considering all reviews.

The outcome of this research is a proposed framework which, first, identify spammer and spam reviews using SGD method and then apply DSR method to produce a diversified set of top-k non-spam reviews. The framework helps in identifying group spammers and displays identified non-spam reviews in diversified manner. To the best of the researchers’ knowledge, this is the first study of its kind to identify spammer group and display diversified set of top-k non-spam reviews about products and services.

Main contributions of the study are as follows:Formulated a co-reviewer graph to identify the connection between different reviewers.

Utilized the SCAN algorithm to identify candidate spam groups.

Employed different individual and group spammer behavioral features to calculate the spam score of each group.

Utilized four different deep learning classifier such as CNN, LSTM, LRU and BERT for training and testing the proposed SGD method.

Proposed a novel diversification method (DSR) to obtain a diversified set of reviews.

The rest of the paper is organized as follows. Introduction section is followed by “Literature Review” which presents related work regarding group spam detection and existing diversification techniques. “Review Datasets” describes the statistics of the datasets used for this research. “Proposed Methods” elaborates the proposed SGD and DSR methods. “Results and Discussion” describes the experimental evaluation of the proposed methods. Finally, “Conclusion” concludes the work.

Literature Review

This study has reviewed the literature from two perspectives (a) identifying the spammer group in the Spam Review Detection (SRD) domain and (b) diversification techniques used in the domain of information retrieval. The aim is to determine the novel contributions of the proposed work by comparing it with prior studies.

Spammer group detection method

In this subsection, existing studies of group spam detection have been reviewed and analyzed. Mukherjee, Liu & Glance (2012) conducted the first study for detecting a group of spam reviewers working together. They used the frequent itemset mining method to get candidate groups and proposed GSRank framework for identifying the spam groups.

Allahbakhsh et al. (2013) also used frequent item mining techniques. Spammer behavioral features like review time and rating scores were used to detect group spammers. They used the Linear Discriminant Analysis (LDA) model by boosting the count of malicious reviewers based on the burstiness of reviews and rating scores. A Spammer Group Detection (PSGD) method was introduced by Zhang, Wu & Cao (2017), which used a supervised learning approach for spotting spammer groups in online review systems. They used frequent item mining to get candidate spammer groups. Then, the Naive Bayesian and Expectation-Maximum (EM) algorithms were used for classification and identification of spammer groups. They performed their experiment on Amazon.cn dataset.

Zhou, Liu & Zhang (2018) identified spammer groups by using self-similarity and clustering coefficient methods. They performed their experiments on Dianping dataset and observed that the clustering coefficient has the best indicator for detecting spammer groups. Rayana & Akoglu (2015) proposed a framework called SPEAGLE which used metadata (review text and spammer behavior) of reviews and relational data (review network). This framework can identify fake reviews, spammers, and target products. They also introduced a lightweight version of SPEAGLE called SPLITE which used a subset of features to avoid computational overhead.

Li et al. (2017) proposed an algorithm to detect individual and group spammers. They proposed Labelled Hidden Markov Modal (LHMM) to identify spammers. They extended their modal to Coupled Hidden Markov Modal (CHMM), which has two parallel HMMs. It represented posting behavior and co-bursting signals. They used hidden states to make a co-bursting network of reviewers to detect spammers who work in a group. Kaghazgaran, Caverlee & Squicciarini (2018) proposed a framework called TwoFace using a neighborhood-based method to spot spammer groups in an online review system. First, they exploited different crowdsourcing websites and selected Rapid Workers to get information about their activities of Amazon products in which they were targeted. Next, they have identified product IDs from the amazon dataset for products mentioned in crowdsourcing activities. Later, they get a list of reviewers who have written reviews about these products and found reviews of all such reviewers who write a review on those products. After that, they have identified all those reviewers who have written reviews on the same product and created a co-reviewer graph. The model, then, applied the trust rank algorithm, which is based on the PageRank algorithm, to find ranking scores of different suspicious groups. They used a different machine learning algorithm to classify suspicious spammer groups.

Zhang et al. (2018) proposed a CONSGD method that used a cosine pattern and heterogeneous information network method to detect spammer groups. To find a tight spammer group candidate, they used the FP-Growth-like algorithm to find cosine patterns. They restricted the tightness of extracted groups with a low cosine threshold value to achieve efficiency. Xu & Zhang (2016) proposed a statistical model called Latent Collusion Model (LCM) to detect spammer groups. This model can identify candidate spammer groups. Xu et al. (2019) proposed a three-phase method called Group Spam Clique Percolation Method (GSCPM) which is based on the Clique Percolation Method (CPM). It is a graph-based method, which models review data as a reviewer graph then breaks this reviewer graph into k-clique clusters using CPM. After that, it ranks these groups using group-based and individual-based spam indicators. The model marked top-ranked groups as spammer groups. In a similar context, Hu et al. (2019) used the CPM method to find spammer groups with the infinite change in the review stream.

Considering the existing work on spam group detection, most of the related studies (Mukherjee, Liu & Glance, 2012; Allahbakhsh et al., 2013; Zhang, Wu & Cao, 2017; Zhou, Liu & Zhang, 2018) have used spammer behavioral features to detect spam groups. On the other hand, some researchers used graph-based techniques to identify suspicious spammer groups with a little focus on spammer behavioral features (Rayana & Akoglu, 2015; Li et al., 2017; Kaghazgaran, Caverlee & Squicciarini, 2018; Zhang et al., 2018; Xu & Zhang, 2016; Xu et al., 2019; Hu et al., 2019). This research aims to develop a framework that will use both behavioral and graph features. First, it creates connections between suspicious reviewers based on the similarity of their behavioral features and then scans the identified suspicious reviewers to detect spammer groups.

Diversification method

This subsection analyzes existing diversification methods used for information retrieval. The first study, about the recommender system using diversification technique, was introduced by Ziegler et al. (2005). They considered top-N reviews and proposed an approach based on clustering, which selects a small subset of reviews that cover better-diversified opinions and high-quality attributes. However, this method used a limited number of reviews, so it is difficult to assure that all required sentiments were considered. Naveed, Gottron & Staab (2013) proposed FREuD method which is based on latent topics. The limitation of the proposed method was that it assigned equal weightage to both the negative and positive sentiment.

Guzman, Aly & Bruegge (2015) applied different weights to the sentiments and allowed stakeholders to assign desired importance to the sentiments. They proposed a diverse method, which retrieved a set of reviews that represented the diversified opinions of users. Moreover, they have also grouped reviews with similar attributes and sentiments.

Naveed, Gottron & Rauf (2018) used probabilistic topic modeling for diversification. They extracted the features of product reviews and displayed the diversified reviews based on these extracted features.

Based on the reviewed literature, it has been observed there exist very limited studies which considered review diversification problem out of which most of the related studies (Ziegler et al., 2005; Naveed, Gottron & Staab, 2013; Guzman, Aly & Bruegge, 2015) have selected diversified set of top-k reviews having positive and negative sentiments based on search query. On the other hand, the study (Naveed, Gottron & Rauf, 2018) used features based approach for displaying top-k reviews using search query. However, these existing studies either identifies sentiments or product feature using search queries and no existing study combined product features and sentiments to display diversified review without considering search queries. The aim of this study is to develop a method which can display reviews in a diversified manner such that the presented reviews represent positive, negative and neutral sentiments covering all related features about product and services. To obtain this objective, this study proposed a novel diversification method (DSR) to display diversified set of reviews using sentiment analysis and product features.

Review Datasets

This study has utilized two datasets: (a) Yelp (spam and non-spam reviews) real-world dataset about hotels and restaurants which was provided by Rayana & Akoglu (2015). Table 1 presents the summary of Yelp dataset and, (b) Roman Urdu product reviews real-world dataset, which was scrapped from Daraz (www.daraz.pk) using python library (Scrappy). Table 2 presents the summary of the Daraz dataset.

Table 1 Summary of Yelp review dataset.

It shows the statistics about Yelp dataset used in this study.

Location	Reviews	Reviewers	Hotels and Restaurants	
YelpChi	67,395	16,063	105	
YelpNYC	96,522	20,225	431	
YelpZip	191,293	38,158	1,952	
Total	355,210	74,446	2,488	

Table 2 Detailed distribution of the Daraz dataset used in proposed methods.

The statistics about Daraz dataset used in this study are shown.

Category	Reviews	Reviewers	Products	
Phones & Tablets	440	149	30	
Clothing & Fashion	370	50	24	
Beauty & Health	270	48	22	
Appliances	390	75	14	
Computers & Gaming	840	206	42	
TV, Audio & Cameras	890	65	37	
Sports & Travel	723	262	48	
Total	3,923	855	217	

This study has utilized Daraz dataset which contains reviews of products from a time span of February 2016 to January 2020. Further, this study also used Yelp dataset containing reviews about hotels and restaurants spanning from March 2011 to August 2014. For this study, we have removed those reviewers who have posted less than two reviews. We have also removed those products which have less than three reviews from the Daraz and Yelp dataset. This study also removed junk characters, numerical values and stop words from review text and removed those reviews which contained less than five words. After pre-processing, the Daraz dataset is reduced to 3,923, reviews and Yelp dataset is reduced to 355,210.

Proposed methods

This research proposed two methods: (i) Spammer Group Detection (SGD) method which detects suspicious groups of reviewers, who write spam reviews to promote or demote the targeted products and services, (ii) Diversified Set of Reviews (DSR) method which selects a diversified set of top-k non-spam reviews having positive, negative, and neutral sentiments. Furthermore, it covers all possible features about the product or service.

Proposed spammer group detection (SGD) method

This section explains the proposed Spammer Group Detection (SGD) method. The framework of the proposed spam review detection method is described in Figure 2. The execution of the SGD starts with Daraz (Roman Urdu reviews) dataset. The proposed framework is divided into three phases. In the first phase, the co-reviewer graph of suspicious reviewers is generated which is based on identified similar behavioral features. The advantage of a co-reviewer graph is that it will create the linkage between the suspicious reviewers which are appeared to be similar based on products reviewed. In the second phase, Structural Clustering Algorithm for Networks (SCAN) utilizes a clustering approach to identify candidate spammer groups. In the third phase, the spam score is calculated for these groups based on individual and group spammer behavioral features. The groups having spam score less than a specific threshold are dropped from candidate spam groups. Moreover, it has been assumed that all the reviews of the identified spam group are considered as spam review.

Figure 2 Framework of proposed Spammer Group Detection (SGD) method.

Co-reviewer graph

In this section, the procedure to generate the Co-Reviewer graph is presented. It is represented by Graph G = (V, E) where vertices V represent the reviewers and edges E represent the similarity between two reviewers. For edge E, a value of 0 represents no similarity between reviewers whereas the value of 1 means that two reviewers are similar. The value of edge E between two reviewers A and B is calculated using Eq. (1) as follows:

(1) λ(A,B)={0,∀p∈PA∩PB,CRS(A,B,p)=01,otherwise

where λ(A,B) represents the weight edge (0,1) between reviewer A and reviewer B. PA and PB are the product sets reviewed by reviewers A and B. Co-Review Similarity between reviewers A and B is represented by CRS(A,B,p). It represents a measure of similarity between two reviewers who are reviewing a product p. It is calculated using Eq. (2) as follows:

(2) CRS(A,B,p)={0,(|tAp−tBp|>α)OR(|RAp−RBp|≥β)1,otherwise

where tAp represents the time when reviewer A reviewed product p whereas tBp represents the time when reviewer B reviewed the product p. α is the user-defined threshold of review time. Through experimental evaluations and analysis of the dataset, it has been observed that most spammers are active for a short period and post their reviews for at most first few couple of days. This is because spammers want to write spam reviews as soon as possible to give positive impact of the product. The researchers have experimentally evaluated different threshold values of α. When threshold value has been increased by 2 then the number of similar reviewers is considerably increased. This also includes those reviewers which are probably the real users and not spammers and have written only one or two reviews. When the threshold value of 1 was used, hardly any similar reviewers were found. Therefore, the optimal value of α is specified as 2-days.

The threshold value β represents reviewer’s rating score which is defined as 2 based on the observations of existing study by Rayana & Akoglu (2015). RAp is the rating given by reviewer A on product p and RBp is the rating score given by reviewer B on product p. Spammer group either tries to promote or demote the product and therefore give an extreme rating which seems different than other reviewers. If they are promoting a product their goal will be to assign a rating of 4 or 5 whereas in the case of defaming a product, they will give 1 or 2-star rating to the product. This is the reason reviewers with a rating difference of 2 or more are not considered as part of any spammer group. If the threshold value of β is set to more than 2, then two reviewers who have rated a product 2 and 5 stars respectively will be treated as similar reviewers which is not true at all. On the other hand, if the rating difference is decreased from 2 i.e., β is set to less than 2, then a large portion of candidate spammer groups will be created. Therefore, the optimal value of β is set to 2.

The calculation of Co-Review Similarity (CRS) is performed based on the following conditions. First, the difference in review posting time of two reviewers is compared and then the difference of the reviewer rating is compared. If review posting time is larger than a user-defined threshold α or the difference of reviewers rating is greater than threshold β, then a value of 0 is given to CRS which represents no similarity between two reviewers. On the other hand, if these two conditions become false then a value of 1 is given to CRS which shows that these reviewers are similar to each other and are behaving almost the same way.

Candidate spam groups using SCAN algorithm

In this section, candidate spam groups are generated from a co-reviewer graph using the Structural Clustering Algorithm for Networks (SCAN) algorithm. SCAN algorithm was proposed by Xu et al. (2007) for identifying clusters, hubs, and outliers in datasets. However, this research has utilized a SCAN Algorithm (Fig. 3) for identifying only clusters where these clusters are represented as candidate spam groups.

Figure 3 Process of structural clustering algorithm for network.

A vertex v∈V, representing a reviewer, is called a core if it has a similar structure (commonly reviewed products in this case) with at least n vertices (or n reviewers) in its neighborhood. Structural similarity of a vertex v with its neighbor x is calculated using Eq. (3) as follows:

(3) StructuralSimiliary(v,x)=no.ofsharedneighborsbetweenvandx(no. of v′s neighbors)×(no.of x′s neigbors)

This structural similarity score identifies the similarity structure of two vertices with each other. The higher the structural similarity score is, the more similar structure these two vertices have and vice versa. For a vertex to be considered as the core, the structural similarity score must be greater than a specified threshold γ with a minimum n number of neighbors. After experimental evaluation, the value of γ is specified as 0.5 and the value of n is taken as 3. A similarity score γ = 0.5 was set to have confidence that both the reviewers have reviewed at-least half of the similar products. For example, if a reviewer has reviewed 4 products whereas the other has reviewed only 1 of those, then both of these should not be treated as spammer-neighbors. For this reason, the value of γ cannot be considered as less than 0.5 because it may include many such instances. On the other hand, when the value of γ was increased from 0.5, only a few such neighbors were found who reviewed similar products. Similarly, the count of minimum neighbor is set to 3. The reason is that if we consider only 1 or 2 neighbors, almost all the reviewers will be included in the spammer group which does not exist in real scenario as there are always some genuine reviewers of the product which should not be treated like spammers. On the other hand, if n is increased from 3, then sparse spammer groups will be identified. This is the reason, a vertex v can be treated as a core if it has a similarity score higher than γ = 0.5 with at least n = 3 neighbors. If a vertex x∈V is in the neighborhood of a core v, then it is called Direct Reachable Structure (DRS). Moreover, the connection of vertices v and x has value 1 as computed in the co-reviewer graph.

Figure 3 elaborates the working of Structural Clustering Algorithm for Networks (SCAN). First, all the vertices are labeled as unclassified (Line 1). SCAN Algorithm classifies these vertices into members or non-members. For every unclassified vertex (Line 2), it is checked that if it is a core (Line 3), if yes, then a new cluster-ID is generated (Line 4). Once a cluster-ID is generated based on this identified core, all the neighbors of the core are inserted into a queue “Q” (Line 5). After inserting all the neighbors of core in Q (Line 6), every element in Q is used to explore Directly Reachable Structure (DRS) vertices from it. These identified DRS are, then, placed in “R” (Line 7). Thereafter, each vertex in R is checked for having neighbors with a structural similarity score greater than a specified threshold γ which are still not assigned to any other cluster (Line 9). Such neighbors of x are inserted into Q with the intention that this cluster can grow from those neighbors (Line 10). Then the classification status of x is checked and if it is unclassified yet, the current cluster-ID is assigned to it (Line 12). In case a vertex is not identified as core by Line 3 then it is labeled as a non-member (Lines 14–15) so that it should not be checked again for being the core. This is specified to minimize the time complexity.

Spam score using spammer behavioral features

This section describes the third phase which calculates the spam score of candidate spam groups as generated by the SCAN Algorithm (Fig. 3). The spam score of every candidate group is calculated using various individual spammer and group spammer behavioral features. The values of these behavioral features are calculated independently and then the average score of these behavioral features is assigned to that candidate group. In this research, a total of eight spammer behavior features (the combination of individual and group spammer behavioral features) are used to assign a spam score to every group. In addition to these spammer behavioral features, a loss function is also used to reduce the contingency of small groups (Wang et al., 2016). The loss function is defined in Eq. (4) as follows:

(4) L(g)=11+e−(|Rg|+|Pg|−3)

where Rg is the number of reviewers in the group and Pg represents the number of products in the group.Individual spammer behavioral features

In this work, three individual spammer behavioral features are used. Table 3 represents the list of notations used in describing individual spammer behavioral features.

Table 3 List of notations used in SGD method.

The symbols used for SGD method are shown.

Notation	Description	
λ(A,B)	Weight edge between reviewers A and B	
CRS(A,B,p)	The co-review similarity between reviewers A and B while reviewing product p	
tAp	The time when reviewer A reviewed product p	
RAp	Rating is given by reviewer A on product p	
L(g)	The loss function for the group g	
Rg	Number of reviewers in group g	
Pg	Number of products in group g	
L(r)	Date of latest review by reviewer r	
F(r)	Date of the first review by reviewer r	
Vr	Number of reviews by reviewer r	
γrp	Rating score is given by reviewer r on product p	
γrp¯	The average rating score of product p (given by all reviewers)	
∩r∈RgPr	Number of common products reviewed among all the reviewers r in group g	
∪r∈RgPr	Number of total products reviewed by all the reviewers r in group g	
S2(p,g)	The variance of rating scores of the product p by reviewers in group g	
Rgp	Number of reviewers who reviewed the product p in group g	

Time burstiness

Usually, spammers target a product in a short period to achieve their goals. Time Burstiness (BST) of a reviewer r is defined in Eq. (5) as follows:

(5) BST(r)={0,L(r)−F(r)>σ1−L(r)−F(r)β,otherwise

where L(r) is the date of the latest review by r, F(r) is the date of the first review by r and σ is the user-defined time threshold specified as 3 days. The spammers while reviewing a product, are generally active for a shorter span of time and once their goal is achieved, they do not generate reviews for the product. If the threshold value is increased from 3, then many real users are also included in the candidate-spammers whereas, decreasing this value from 3 returned very few spam reviewers.

Maximum number of reviews

Generally, spammers tend to post a larger number of reviewers in a single day. Maximum number of reviews (MNR) for a reviewer r is defined in Eq. (6) as follows:

(6) MNR(r)=maxVrmaxr∈R(maxVr)

where Vr is the number of reviews posted by r in a day and it is normalized by the maximum number of reviews in the reviewer’s review set.

Average rating deviation

Mostly, a spammer gives a different rating from the genuine reviewer’s ratings because the purpose of the spammer is a false projection of a product either in a positive or negative sense. Average rating deviation (ARD) is defined in Eq. (7) as follows:

(7) ARD(r)=avgp∈Prδrp−δrp¯5

where Pr is the set of products reviewed by reviewer r, δrp represents rating score given by r to the product p and δrp¯ represents the average rating score of product p given by all reviewers. This value is then normalized by the maximum rating deviation i.e., 5 in case of a 5-star rating system.Groups spammer behavioral features

A total of five group spammer behavioral features are used in this work. The symbols used in describing group spammer behavioral features is represented in Table 3.

Review tightness

Review tightness (RT) of a group is defined as the similarity of reviews by the reviewers of candidate spammer group. It is defined in Eq. (8) as follows:

(8) RT(g)=|Vg||Rg||Pg|

where |Vg| represents the number of reviewers in group g whereas |Rg||Pg| is the cardinality of the Cartesian Product of the reviewer set and the product set in group g.

Product tightness

Generally, in a spam group, the spammers target some specific products therefore the product tightness (PT) is an important spammer behavioral feature. It represents the similarity of products reviewed by reviewers of candidate spammer group. It is defined in Eq. (9) as follows:

(9) PT(g)=|∩r∈RgPr||∪r∈RgPr|

where |∩r∈RgPr| represents the number of the common product reviewed by the members in the group and |∪r∈RgPr| represents all products reviewed by all members of the group.

Rating variance

The members of a candidate spammer group usually give a similar rating to the reviewed products. This type of spammer behavior can be identified by calculating Rating Variance (RV) which is defined in Eq. (10) as follows:

(10) RV(g)=2(1−11+e−avgp∈PgS2(p,g))L(g)

where S2(p,g) represents the variance of the rating scores of product p by all reviewers of group g.

Group size

It has been observed by existing study (Mukherjee, Liu & Glance, 2012) that usually spammer groups of 2 or 3 reviewers are formed by coincidence who have no intensions or common interest with each other to write spam reviews. However, larger groups are usually formed by the intention to write spam reviews to target any product or service. Therefore, identifying group size (GS) is a good feature to observe the behavior of the candidate spammer groups. Moreover, it is worthwhile to give more weightage to the larger group size. The group size indicator is defined in Eq. (11) as follows:

(11) GS(g)=11+e−(|Rg|−3)

where Rg is the number of reviewers in group g.

Reviewer ratio

In a candidate spammer group, if some products are being reviewed by one reviewer while other reviewers of the same spammer group have not posted any reviews about that product than this represents suspicious behavior of the reviewers of the candidate spammer group. The Reviewer Ratio (RR) is therefore calculated to assess this behavior of candidate spammer group. It is represented by Eq. (12) as follows:

(12) RR(g)=maxp∈Pg|Rgp||Rp|

where Rgp represents the number of reviewers in group g who reviewed the product p and Rg is the total number of reviewers who reviewed the product p.

Based on the calculated values of these behavioral features (Eqs. (5)–(12)), the spam score of each candidate spammer group is calculated by taking an average score of these eight behavioral features. This spam score highlights the suspiciousness of candidate spammer groups such that the higher the spam score, the more chances are for that group having spam reviewers. Through experimental evaluations, a threshold value of 0.6 is defined for spam score which is used to identify suspicious spammer groups. While analyzing the existing datasets, it was observed that generally, 10–15% of total reviews are spam so this study decided to follow the same ratio. When this threshold was set to 0.4–0.5 all the candidate-groups were treated as spam that resulted in almost 40% of the reviews as spam. On the other hand, increasing this value to greater than 0.6 resulted in only a few candidate spam groups, which produced less than 5% of the total reviews as spam. Therefore, the spam score threshold was set to 0.6 which labeled 13.51% of reviews as spam and provided optimal results.

It can also be assumed that the threshold value can vary depending upon different applications. For example, when an application wants to identify as many spam reviews as possible, then he or she ought to set threshold value to be relatively small. After identifying suspicious spammer groups, all the reviews by the members of these groups are labeled as spam which results in the labeled dataset.

Proposed diversified set of reviews (DSR) method

In this section, a novel DSR method is proposed which returns a compact set of diversified non-spam reviews having positive, negative, and neutral sentiments covering the maximum number of features. In contrast to the earlier techniques, this work does not only retrieve the reviews based on diversified sentiments (i.e., positive, negative, and neutral) but also displays reviews covering all possible product features. Product features represents important components of the product about which customers are writing reviews. For example, consider laptop is a product; its features will represent its battery life, display screen, RAM, and performance etc. The proposed DSR approach displays diversified reviews based on the features and sentiments simultaneously. Figure 4 represents the framework of the DSR method which shows that the review dataset (not-spam reviews), achieved through the SGD method, is utilized as an input of DSR. In this approach, first, this review dataset of a specific product is divided into three categories based on positive, negative and neutral sentiments, then, the diversified feature extraction process is applied on these categories separately. In feature extraction process, feature set is prepared for each product review. For example, consider a product review “When I received the device and hit the power button, it didn’t turn on easily. After setting it up, I notice that the volume up key doesn't work and the volume was raised randomly”. In this review “power button” and “volume up key” are features. Similarly, feature set of all reviews of the all products are prepared. For a specific product, three feature sets are prepared based on its positive, negative and neutral categories. Next, each review of a product is assigned a weight based on its features using Eq. (13) which is then used to calculate its utility score. All reviews are evaluated by comparing their utility scores to select top-k reviews of the product. Figure 5 explains this process in detail. Finally, all three diversified datasets of a product (positive, negative and neutral) are combined to display top-k diversified reviews having varied sentiments and expanded product features. The sentiments and features extraction for the reviews are determined by using python’s built-in library TextBlob (Loria, 2018).

Figure 4 Framework of Diversified Set of Reviews (DSR) method.

It explains the proposed Diversified Set of Reviews (DSR) method.

Figure 5 Procedure to find a diversified set of reviews.

Figure 5 elaborates the working of DSR algorithm, which begins with the set of reviews having positive, negative, and neutral reviews (Line 1). The diversified reviews result set S is initially set to empty (Line 2). The algorithm considers one type of sentiment (positive, negative, or neutral) at a time and iterates over all three sentiments (Line 3). For example, if this study has 25 positive, 15 negative and 20 neutral reviews in dataset R, then in the first iteration the value of k will consider all positive reviews (line 4), in the next iteration it will consider all negative reviews and in the last iteration, all neutral reviews are considered. In the next step, the feature set F is formulated which consists of all features described by review r (Line 5). The diversified set for every sentiment is selected and retrieved separately and is stored in set s (Line 6). The loop iterates through the count of specific sentiment k (Line 7). For instance, if the number of positive reviews to be returned is 25, this loop will run 25 times, and in every iteration, it will retrieve one review to be added into the set s. For each review (Line 8), the addressed features of reviews are observed. If the same feature exists in feature list F, then these are added into a list f∗(r) (Line 9). To maximize the diversification process, the features selected from set F for review ri are not considered again for the next review rj.

The weights for these features are calculated using Eq. (13) as follows:

(13) w(f)=c(f)maxf′∈Fc(f′)

where w(f) is the weight of a feature, c(f) represents the frequency of feature f (whose weight is to be calculated) in set F whereas maxf′∈Fc(f′) is the highest frequency of any feature f′ available in the feature list F. These calculated weights of the features are then summed up as Utility (U) of the review r (Line 10). After calculating utility score for each review, review r with the maximum utility is added into s (Lines 12–13) and the same is discarded from the review dataset R subsequently (Line 14). Moreover, the features addressed in r are also eliminated from the feature list F (Line 15) with the aim that these features may not be added in the utility of any other review to achieve maximized diversity. This updates the feature list after every selection. The advantage of updating the feature list is that the remaining unaddressed features are also considered to be displayed in a top-k diversified set of reviews. This feature set is regenerated for every sentiment-based category of dataset i.e., positive, negative, and neutral. Once a sub-set s for a specific sentiment is retrieved, it is appended into original diversified set S (Line 17). This diversified set of reviews are returned and presented to the end-user as top-k diversified review which consists of all positive, negative, and neutral reviews covering all possible product features.

Figure 6 represents the main contribution and framework of this study which identifies spammer groups and presents these non-spam reviews in diversified format. The execution of the framework starts with Yelp and Daraz datasets. Daraz dataset is initially unlabeled. The proposed Spammer Group Detection (SGD) method is used to label the Daraz dataset and highlight the spammer and spam reviews. Yelp dataset is already labelled thus SGD method has not been applied on it. The complete description of the datasets has been described in “Review Datasets”. The working of the SGD method has been described in “Proposed Spammer Group Detection (SGD) Method”. Next, the labelled datasets are fed into deep learning classifiers for training and testing. The output of deep learning models is non-spam reviews from the Yelp and Daraz datasets. These non-spam reviews are considered for Diversified Set of Reviews (DSR) method. The complete working of DSR method has been described in “Proposed Diversified Set of Reviews (DSR) Method”. The output of the DSR method is to display diversified set of top-k non-spam reviews having positive, negative, and neutral reviews/feedback covering all possible product features.

Figure 6 The framework of the proposed study using SGD and DRS methods.

Results and Discussion

This study is evaluated in the following two perspectives: (i) Evaluation of proposed Spammer Group Detection (SGD) Method using four deep learning classifiers (CNN, LSTM, GRU and BERT) in terms of accuracy in spam review detection. (ii) Evaluation of proposed Diversified Set of Reviews (DSR) method in terms of a diversified set of reviews. These evaluation results have been presented in the following subsections.

Evaluation of spammer group detection (SGD) method using deep learning classifiers

This section describes an evaluation of the proposed SGD method which identifies suspicious spammer groups and spam reviews utilizing deep learning classifiers. It presents the analysis of different parameter settings to find out optimal parameters that can be used in deep learning classifiers for the training and testing of the SGD method. The study has used standard evaluation measures (Hussain et al., 2019) to analyze the performance of the proposed SGD method. These include precision, recall, F1score and accuracy. Deep learning classifiers which are used for training and testing of the proposed SGD method are LSTM, GRU, CNN and BERT. In addition to it, K-fold cross-validation (k = 5) is used to validate the accuracy of the proposed method. The datasets (Daraz and Yelp) are split in the ratio of 80 to 20 for training and testing so that more datasets can be utilized to train deep learning classifiers (Hajek, Barushka & Munk, 2020). The experimental evaluation of SGD is performed in three phases: (i) In the first phase, analysis of different parameter settings has been performed to achieve optimized hyperparameters for the proposed deep learning-based SGD method. (ii) In the second phase, the SGD method has been evaluated using different deep learning classifiers to analyze its accuracy. (iii) Finally, the performance comparison of the proposed SGD method has been conducted with existing approaches using different datasets.

Analysis of hyperparameters

In the first set of experiments, several initial parameters have been considered, as represented in Table 4. These initial parameters are used as a starting point of analysis to find out optimized hyperparameters of deep learning classifiers using Daraz and Yelp datasets. Table 4 represents the values of these initial parameters which are used in the analysis of LSTM, GRU, CNN and BERT deep learning classifiers.

Table 4 Initial parameters used for deep learning classifiers.

Parameter	CNN	LSTM	GRU	BERT	
Dropout Rate	0.5	0.5	0.5	0.5	
Optimizer	Adam	RMS Prop	RMS Prop	Adam	
Activation Function	Relu	Relu	Relu	Relu	
No. of Features	1,000	1,000	1,000	1,000	
No. of Units	N/A	128	100	100	
Filter Size	4	N/A	N/A	N/A	

The analysis of deep learning classifiers requires numerical values as an input; therefore, the review text needs to be converted into a numerical form (Vidanagama, Silva & Karunananda, 2020; Lee, Kim & Song, 2020). In this study, Daraz review dataset is converted into the numeric form using Term Frequency-Inverse Document Frequency (TF-IDF) vectorization, whereas, Yelp review dataset is initialized by finding the corresponding word embedding using Google’s Word2vec tool (https://code.google.com/archive/p/word2vec) which used the dimension of 300. The following subsections describe the comprehensive experimental details of different parameters which helped in optimizing deep learning classifiers.

Activation function

The activation function takes the output signal from the previous node (layer) and converts it into some usable form that can be taken as input to the next node (layer). This study, first, analyzes different non-linear activation functions (tanh, relu and sigmoid) on different deep learning classifiers. Next, based on the experimental results, the best performing activation function is utilized in deep learning classifiers for training and testing of the proposed SGD method. Figure 7A presents the experimental results of different activation functions applied to deep learning classifiers utilizing Daraz and Yelp datasets.

Figure 7 Different activation functions.

(A) Effect of different activation functions on the accuracy of deep learning classifiers utilizing Daraz and Yelp datasets. (B) Activation functions utilizing Daraz dataset. (C) Activation functions utilizing Yelp dataset.

It has been observed from Fig. 7B that on Daraz dataset, sigmoid function performs better for CNN and GRU classifiers whereas, relu performs better for LSTM classifier and tanh performs better for BERT classifier. It has also been observed from Fig. 7C that on Yelp dataset, the sigmoid function performs better for LSTM, GRU and LSTM classifiers while, tanh performs better for BERT classifier. Therefore, this study utilized best performing activation function in a deep learning classifier to obtain an input signal for the next node using the output of the previous node.

Optimization method

Deep learning classifiers usually have a certain loss or error during training (Moraes, Valiati & Gavião Neto, 2013). This loss or error is calculated using a cost function. The purpose of the optimization function is to effectively train the classifier such that the error or loss is minimized. This study analyzes different optimization methods (SGD, RMSProp, Adagrad, Adadelta, Adam, Adamax and Nadam) on different deep learning classifiers. Based on the experimental results, the best performing optimization method is utilized in deep learning classifiers for training and testing of the proposed SGD method. Figure 8A presents the experimental results of different optimization methods applied to deep learning classifiers utilizing Daraz and Yelp datasets. It has been observed from Fig. 8B that on Daraz dataset, Adam performs better for LSTM, GRU and BERT classifiers whereas, Nadam performs better for CNN classifier. It has also been observed from Fig. 8C that on Yelp dataset, the Adamax optimization method performs better for LSTM and GRU classifiers while, RMSProp performs better for CNN classifier and SGD performs better for BERT classifier. This study utilized best performing optimization function in deep learning classifiers to effectively train the model such that the error or loss is minimized.

Figure 8 Different optimization methods.

(A) Effect of different optimization methods on the accuracy of deep learning classifiers utilizing the Daraz and Yelp datasets. (B) Optimization methods utilizing the Daraz dataset. (C) Optimization methods utilizing the Yelp dataset.

Dropout function

Deep learning classifiers usually have an overfitting problem, especially on a low volume of data (Wu et al., 2020). Therefore, the dropout function is used to overcome the overfitting problem. Figure 9A presents the experimental results of different dropout rates applied to deep learning classifiers utilizing Daraz and Yelp datasets. It has been observed from Fig. 9B that on Daraz and Yelp (Fig. 9C) datasets, dropout values between (0.2 to 0.5) tends to show good results. Therefore, this study utilized this dropout rate (0.2 to 0.5) in deep learning classifiers to effectively handle the overfitting problem.

Figure 9 Different dropout values.

(A) Effect of different dropout values on accuracy of deep learning classifiers using the Daraz and Yelp datasets. (B) Dropout values utilizing the Daraz dataset. (C) Dropout values utilizing the Yelp dataset.

Number of units

The architecture of the deep learning classifiers is generally controlled by the number of units (layers) and the number of nodes in each hidden unit (Pandey & Rajpoot, 2019). Figure 10A presents the experimental results of adapting different number of units (50, 100, 150 and 200) in deep learning classifiers utilizing Daraz and Yelp dataset. Through experimental evaluations, no significant change has been observed after adding several units as presented in Figs. 10B and 10C.

Figure 10 Different number of units.

(A) Effect of the number of units on the accuracy of deep learning classifiers using the Daraz and Yelp datasets. (B) Number of units utilizing the Daraz dataset. (C) Number of units utilizing the Yelp dataset.

Number of features

In deep learning classifiers words in the datasets are mostly represented as features (Deng et al., 2019). The number of features to be fed in the classifier need to be limited to the most frequently occurring words rather than taking all the features. This helps to reduce the overfitting problem. Figure 11A presents the experimental results of utilizing several features (1,000, 2,000, 3,000 and 4,000) on deep learning classifiers using Daraz and Yelp datasets. It has been observed from Fig. 11B that on Daraz dataset feature set of 3,000–4,000 words performed better for LSTM, GRU and BERT classifiers. On the other hand, on Yelp dataset (Fig. 11C), feature set of 2,000 words performed better for LSTM and GRU classifiers and 4,000 words performed better for BERT classifier. Through experimental evaluations on CNN classifier, it is observed that applying hyperparameter (number of features) on CNN decreases its accuracy value. Based on this analysis, this study utilized best performing feature set (highlighted in table of Fig. 11) in deep learning classifiers to overcome the overfitting problem.

Figure 11 Different number of features.

(A) Effect of the number of features on the accuracy of deep learning classifiers utilizing the Daraz and Yelp datasets. (B) Number of features utilizing the Daraz dataset. (C) Number of features utilizing the Yelp dataset.

Optimized parameters for deep learning classifiers

A comprehensive experimental evaluation is presented in “Analysis of Hyperparameters” to find out optimized hyperparameters of deep learning classifiers. After performing the analysis of different parameter settings, the final set of optimized parameters for the proposed deep learning-based SGD method is shown in Table 5. These parameters settings are used for the training and testing of deep learning classifiers (CNN, LSTM, GRU and BERT).

Table 5 Optimized parameters used for the analysis of deep learning classifiers.

Parameters	CNN	LSTM	GRU	BERT	
Dropout Rate	0.3	0.3	0.2	0.3	
Optimizer	Nadam	Adam	Adam	Adam	
Activation Function	Sigmoid	Relu	Sigmoid	tanh	
No. of Features	N/A	2,000	2,000	3,000	
No. of Units	200	50	100	100	
Filter Size	4	N/A	N/A	N/A	

Analysis of deep learning classifiers

In this subsection, deep learning classifiers are evaluated in terms of accuracy achieved by each classifier. Table 6 shows that the CNN classifier performs better than the other two classifiers (LSTM and GRU) on Daraz and Yelp datasets. It has been observed from the literature review (Yin et al., 2017) that CNN may perform better for text classification. This observation can be applied in this study and CNN classifier can perform better because it utilized a review (text) dataset about products and services. CNN classifier uses the kernel which slides along the features and the weights. This mechanism may also be in favor of the utilized datasets. It is observed from the literature review that the LSTM classifier processes the features or words using sequential learning method (Rao et al., 2018). This process may not be in favor of the utilized datasets. Therefore, CNN produces better accuracy as compare to LSTM. It was observed by the literature review that the GRU classifier uses internal memory for storing and filtering information using their update and reset gates (Khadka, Chung & Tumer, 2017; Reyes-Menendez, Saura & Filipe, 2019). Therefore, this feature can produce a better accuracy score as compared to the LSTM classifier. It has been observed from literature review that BERT perform better for the applications where search query is used to display matching results using Natural Language Processing (NLP) and may also best suited when analyzing sentiments (positive or negative) of the reviews (Jacob et al., 2020). These observations cannot be favorable for this study as the framework is based on the identification of group spammer using individual spammer behaviors and group spammer behavior features. Therefore, BERT does not achieve better accuracy score as compare to other classifiers.

Table 6 Performance comparison of deep learning classifiers using Daraz and Yelp datasets.

Deep Learning Classifiers	Daraz Dataset	Yelp Dataset	
Precision	Recall	F1 Score	Accuracy	Precision	Recall	F1 Score	Accuracy	
BERT	71.22	72.41	72.55	73.07	75.87	76.25	77.11	79.89	
LSTM	72.44	73.21	73.88	74.12	80.85	81.02	82.14	83.14	
GRU	73.96	75.55	74.88	76.55	82.14	84.10	85.11	86.21	
CNN	77.15	79.21	78.11	81.31	85.78	87.14	88.47	89.41	

Table 6 shows that deep learning classifiers produce better accuracy results utilizing Yelp dataset (355,210 reviews) as compared to the Daraz dataset (3,923 reviews). The reason for this worst performance on Daraz dataset is that deep learning classifiers use word2vec which utilizes semantics to encode the words. This requires a big corpus so that word2vec can build a good vocabulary for encoding. As the Daraz dataset does not provide a big corpus for Roman Urdu reviews as compared to the Yelp dataset, therefore, in this study deep learning classifiers produce better accuracy results on Yelp dataset as compare to Daraz dataset.

Performance comparison with existing approaches using different datasets

In this subsection, we have presented the comparison of the proposed SGD method with previously available studies which are using various datasets for group spam detection. This comparison has been presented in Table 7 which demonstrates the effectiveness of proposed approach in terms of achieved accuracy using Amazon and Yelp datasets. Amazon dataset is a real-world product review dataset. In order to conduct this comparison, this study utilized 15,342 reviews, 10,522 reviewers and 5,312 products of the Amazon dataset. Yelp dataset contains 355,210 review, 74,446 reviewers of 2,488 hotels and restaurants. The accuracy results presented in Table 7 shows that the proposed Spammer Group Detection (SGD) method has outperformed the existing methods (Mukherjee, Liu & Glance, 2012; Rayana & Akoglu, 2015; Kaghazgaran, Caverlee & Squicciarini, 2018). Mukherjee, Liu & Glance (2012) utilized spammer and group behavioral features to identify spam reviews on Amazon dataset and obtained an accuracy of 86%. Compared to this, the proposed approach achieved an accuracy of 91% when implemented on Amazon dataset. Next, we have compared our proposed approach with Kaghazgaran, Caverlee & Squicciarini (2018) using the Amazon dataset which utilized a neighbourhood-based method to spot spammer groups in an online review system. The proposed SGD method achieves an improved accuracy of 91% as compared to the obtained accuracy of 89% by Kaghazgaran, Caverlee & Squicciarini (2018). Finally, we have performed the comparison of the proposed approach by utilizing linguistic and behavioral features using the Yelp dataset. The results show that the proposed SGD method improved the accuracy to 86% when compared with the approach proposed by Rayana & Akoglu (2015) which achieved an accuracy of 79%. Table 7 presents a comprehensive comparison which validates the improved performance of the proposed approach compared to existing studies utilizing different datasets.

Table 7 Comparative analysis of proposed SGD method with existing approaches using different datasets.

Existing Studies	Dataset	Accuracy (%)	Proposed SGD Accuracy (%)	
Mukherjee, Liu & Glance (2012)	Amazon	86	91	
Kaghazgaran, Caverlee & Squicciarini (2018)	89	91	
Rayana & Akoglu (2015)	Yelp	79	86	

Evaluation of diversified set of reviews (DSR) method

In this section, the proposed DSR method is evaluated in terms of presenting reviews in diversified manner representing positive, negative, and neutral sentiments covering all related features about product and service. The execution of the DSR method works in two phases which takes in non-spam reviews obtained using SGD method from Daraz and Yelp datasets. (1) For the sentimental analysis phase, this study utilizes python’s built-in library TextBlob (https://pypi.org/project/textblob/) for Daraz and Yelp dataset to obtain positive, negative, and neutral sentiments of reviews. (2) In feature extraction phase, features are extracted from the review datasets using two different methods: (i) For Daraz dataset, list of unique features or words are generated by writing programming code and almost 7,344 unique Roman Urdu words or features are considered for further evaluation; (ii) For Yelp dataset, this study utilizes python’s built-in library TextBlob for feature extraction. After evaluating sentimental analysis and feature extraction of both review datasets (Daraz and Yelp) with non-spam review, the Diversified Set of Reviews (DSR) algorithm has been used to present reviews in a diversified manner such that the presented reviews represent positive, negative and neutral sentiments covering all related features about specific product and services. This study initially set-top-k reviews (k = 10) considering four positive, four negative and two neutral reviews. This top-k value can be adjustable according to the requirement of the end-user.

Figure 12 presents the working of the proposed DSR method using real reviews from Yelp dataset. Figure 12A shows non-spam reviews of a specific hotel which is located in New York whereas Fig. 12B presents top-k non-spam reviews having positive, negative, and neutral reviews/feedback covering all possible features about hotel after applying DSR method. For the reader’s convenience, the features of each review are highlighted in bold. In this example top-k value has been set to five whereas it displays two positives reviews, two negative reviews and one neutral review.

Figure 12 Example of Yelp review dataset.

(A) Presentation of reviews before utilizing the DSR method. (B) Presentation of reviews after utilizing the DSR method.

The current research proposed DivScore to analyze the performance of DSR method. The DivScore is calculated on the basis of features addressed in each review. There exists a relation between DivScore and review diversification. The higher value of DivScore represents more diversified top-k reviews. The impact score can be reduced if the feature appears more than one time in the top-k reviews. For example, if the occurrence of the feature in the review dataset is “1” then its overall score is “1” but if its occurrence is three times then the score of the feature will be reduced to 1/3 = 0.33. The impact score is calculated by the following Eq. (14).

(14) ImpactScore(f)=1countoffinS

In above equation, S represents a diversified set of reviews. After calculating the impact score for all the features, the scores for the feature in a review are used to calculate Review Diversity Score (Eq. (15)) for that review. As the review diversity score is higher, more features are being addressed by that review and more diversified that review is from the remaining diversified set of reviews. The mathematical representation for Review Diversity Score is given in Eq. (15).

(15) ReviewDiversityScore(r)=∑f∈r⁡ImpactScore(f)

For calculating DivScore, the review diversity scores (Eq. (15)) are normalized by dividing it with the maximum review diversity score in the review set. In the end, these normalized scores are summed to obtain the DivScore for the diversified set. Equation (16) is used to calculate DivScore.

(16) DivScore=∑r∈SReviewDiversityScore(r)maxr∈S(ReviewDiversityScorer)

Table 8 shows experimental evaluation conducted on the products/services of the Daraz and Yelp datasets. For this evaluation, top ten products and services, having maximum reviews, have been selected from Daraz and Yelp datasets. The reason to select top ten products having maximum reviews is that these can represent maximum features about these products and services. The diversified set obtained for these products achieved a DivScore which has been displayed in the Table 8 for all ten products. It can be observed that the services of Yelp dataset achieved better DivScore than the products of Daraz dataset. The reason behind it is that total reviews per services of Yelp dataset are more in quantity as compare to the total reviews per product of Daraz dataset which produced rich and diversified set of features for analysis.

Table 8 Performance evaluation of proposed DSR on the Daraz and Yelp reviews.

Top Products and Services	Yelp Dataset	Daraz Dataset	
Positive Reviews	Negative Reviews	Neutral Review	Total Reviews	DivScore	Positive Reviews	Negative Reviews	Neutral Review	Total Reviews	DivScore	
1	6,282	274	92	6,648	10.8	41	27	12	80	7.1	
2	4,968	722	128	5,818	9.9	43	14	19	76	6.9	
3	3,714	225	53	3,992	8.8	25	15	8	48	6.5	
4	3,127	221	34	3,382	8.7	24	11	7	42	6.4	
5	2,695	169	76	2,940	8.1	23	13	6	42	6.3	
6	2,596	278	43	2,917	8.1	21	17	2	40	6.2	
7	2,500	261	39	2,800	7.9	25	11	3	39	6.2	
8	2,288	257	43	2,588	7.6	20	10	7	37	6.0	
9	2,378	158	40	2,576	7.6	21	10	5	36	5.8	
10	2,328	137	28	2,493	7.5	20	13	2	35	5.7	

Conclusion

This study proposed Spammer Group Detection (SGD) method and Diversified Set of Reviews (DSR) method to evaluate real-world datasets such as Daraz and Yelp. SGD method used linguistic, behavioral and group spammer behavioral features to calculate the spam score of each group to identify group of the spammers. This study used deep learning classifiers for training and testing the proposed SGD method. The proposed Diversified Set of Reviews (DSR) method utilized diversification technique to present a diversified set of top-k reviews having positive, negative, and neutral feedback covering all possible product features about a specific product or service. The study proposed a framework which works by combining SGD method with DSR method. The output of SGD method which is non-spam reviews are used as input to DSR method. The outcome of this framework are non-spam reviews of a specific product displayed in diversified manner. The findings of this research provide a practical implication for improving the trustworthiness of online products and services for Roman Urdu and English reviews in diversified manner. In future, additional attributes such as the email id, IP address of the spammer and signed-in location of the reviewer may be added to the dataset to improve the accuracy of the spam review detection model. Moreover, another future direction is to include location-dependent behavioral features of reviewer for in-depth identification of spam reviews.

Supplemental Information

Supplemental Information 1 Daraz product labeled review dataset with sentiments and features.

Daraz product reviews dataset with spam and non-spam labels. The dataset also contains sentiments (positive, negative and neutral) and features from review.

The dataset is also available at Kaggle: Naveed Hussain, “Daraz Roman Urdu Reviews.” Kaggle, 2021, doi: 10.34740/KAGGLE/DSV/1898516.

Click here for additional data file.

Supplemental Information 2 Yelp reviews dataset with sentiemnts and features.

The Yelp reviews (spam and non-spam) with sentiments (positive, negative, and neutral) and review features.

The complete dataset is available at Kaggle: Naveed Hussain, “Yelp Review with Sentiments and Features.” Kaggle, 2021, doi: 10.34740/KAGGLE/DSV/1898501.

Click here for additional data file.

Supplemental Information 3 The Code for DSR method using Daraz dataset.

The complete code for analyzing DSR method utilizing Daraz dataset.

Click here for additional data file.

Supplemental Information 4 Code for analyzing the DSR method using the Yelp dataset.

The complete code for analyzing the DSR method utilizing the Yelp dataset.

Click here for additional data file.

Supplemental Information 5 Code for analyzing the SGD method using the Yelp dataset.

The complete code for analyzing the SGD method utilizing the Yelp dataset.

Click here for additional data file.

Supplemental Information 6 Code for analyzing the SGD method using the Daraz dataset.

The complete code for analyzing the SGD method utilizing the Daraz dataset.

Click here for additional data file.

Additional Information and Declarations

Competing Interests

Author Contributions

Data Availability

The authors declare that they have no competing interests.

Naveed Hussain conceived and designed the experiments, prepared figures and/or tables, and approved the final draft.

Hamid Turab Mirza conceived and designed the experiments, prepared figures and/or tables, and approved the final draft.

Abid Ali performed the experiments, authored or reviewed drafts of the paper, and approved the final draft.

Faiza Iqbal performed the computation work, prepared figures and/or tables, proofreading, and approved the final draft.

Ibrar Hussain analyzed the data, authored or reviewed drafts of the paper, and approved the final draft.

Mohammad Kaleem performed the computation work, authored or reviewed drafts of the paper, and approved the final draft.

The following information was supplied regarding data availability:

The Daraz dataset is available at Kaggle: Naveed Hussain, “Daraz Roman Urdu Reviews.” Kaggle, 2021, DOI 10.34740/KAGGLE/DSV/1898516.

The Yelp dataset is available at Kaggle: Naveed Hussain, “Yelp Review with Sentiments and Features.” Kaggle, 2021, DOI 10.34740/KAGGLE/DSV/1898501.

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
