# Peer review of "Spammer group detection and diversification of customers’ reviews"

_PeerJ Computer Science, doi:10.7717/peerj-cs.472_

## Round 0.1 · original submission · Major Revisions

Dear authors,

Reviewers have now commented on your paper. They are advising that you revise your manuscript. If you are prepared to undertake the work required, I would be pleased to reconsider the submission.

Reviewer 1 ·

Basic reporting

The article covers an interesting topic, spam detection, with the use of a trending technique such as deep learning. It provides a good background on references and tries to solve the problem of spam in Non-English languages (Urdu). That being said, I must say that I am starting to get into this research area, and I am not a great expert in this field.

Experimental design

This paper shows a novel technique to detect spam using deep learning techniques. This is a technique that is getting lots of focus lately. However, even though I do know it is hard to find a data repository and on top of that significative data repository in urdu which is really hard, the goal of the study is to detect spammer groups and it has become a greater deal during the last five years (elections, pandemic, shopping, social networks, …) and the data is relatively old to infer that the technique is valid nowadays. For instance, and roughly speaking, we can think of an "Amazon Verified Purchase" to simplify certain computations. But, in general, I would appreciate if you can do or, at least, infer how it should work nowadays.

Finally, and this is only an example, Google has presented BERT and I do not know how it overcomes or facilitate a better detection of spammer groups than the latter. I would appreciate if you can tell something about that. For instance, the use of BERT with your technique, from the hyperparameters to the implementation of both techniques.

Validity of the findings

My main concern with the aim of the paper is the two branches of study. You propose two methods that I can see and understand both, but they seem like two different types of work and I don’t see the connection between these two. For instance, at the beginning of Section 4 I would appreciate a connection between both or in the conclusions. Regarding to the latter, I humbly think it should be rewritten to take into account the usefulness of both techniques working as they are and together. Right now, it has a lot of future work and almost an enumeration of Sections 4 and 5.

Reviewer 2 ·

Basic reporting

This paper presents a study on spam review identification and review diversification. Solving these two tasks will help improve customer experience for making better informed purchasing choices. The paper presented a Spammer group detection (SGD) method, which involves three stages: 1) construct a co-reviewer graph; 2) cluster reviewers using Structural Clustering Algorithm for Networks (SCAN); and 3) identify spammer groups using eight spammer behavioral features. Review diversification was designed to display reviews representing different sentiments and product/service features. Experiments were conducted on two datasets in different languages, English and Urdu.

In the introduction section, the paper gave a nice overview of the SGD method. It would be better if the authors could enrich the description about the review diversification part (line 93-95). Similarly, in literature review, the paper presented existing clustering methods and their comparison with a proposal for review spammer detection. It would be better if a comparison between the proposed review diversification method and existing ones (line 185-189).

Section 4.2 and Figure 4 presented that the review diversification method used features. However, it is unclear what the features were. Were these features word-based features (from looking at the excel sheet provided in the supplemental material)? It is good to see the authors provided an excel sheet showing the features for the Daraz dataset. It would be better if the authors can describe in details how they used the features in the paper.

Experimental design

The paper provided detailed parameter analysis of tuning different deep learning models. However, it is unclear to me why the authors introduced deep learning models for evaluation and how the deep learning models were used to evaluate the SGD method. Were spammer scores described in Section 4.1.3 used as the ground truth labels which were further used for training and testing the deep learning models?

In Section 4.1, SGD needs several parameters, e.g., threshold of review time and threshold of review rating (Equation (2)), minimum number of neighbors (Equation (3)), threshold of review time (Equation (5)), and threshold of spammer group identification (line 358-359). The paper mentioned those parameters were determined through experimental evaluations. However, it is unclear how those parameters were tuned.

For the review diversification evaluation, the results could be more convincing if they are compared with existing methods described in the literature review section similar to what the authors did for spammer group detection comparison.

Please provide the reference of the standard evaluation measure for diversification “divScore” (line 551-552).

Validity of the findings

Thanks for providing the dataset. However, it is unclear how the datasets provided in the supplemental material were used for experiments. There are few places in the paper mentioning the dataset statistics, e.g., line 202-203, line 507-508, line 518-520, and line 566-572. However I found it is difficult to match those statistics against those in the excel sheets provided in the supplemental material. The Yelp dataset for review diversification is missing.

Were the reviews in Figure 1 obtained before and after running your diversification algorithm? If not, could you provide examples before and after you run your review diversification algorithm? How does your algorithm help to present a comprehensive overview of a product or service features?

Additional comments

This paper presents a study on spam review identification and review diversification. The paper presented a Spammer group detection (SGD) method, which involves three stages: 1) construct a co-reviewer graph; 2) cluster reviewers using Structural Clustering Algorithm for Networks (SCAN); and 3) identify spammer groups using eight spammer behavioral features. Review diversification was designed to display reviews representing different sentiments and product/service features. Experiments were conducted on two datasets in different languages, English and Urdu.

Solving these two tasks will help improve customer experience for making better informed purchasing choices. The paper is easy to follow. The paper could be stronger if more descriptions and experiments on the review diversification part, descriptions on parameter setting for SGD, and more descriptions on how much data are used for each experimental settings are provided,.

---

## Round 0.2 · Minor Revisions

Dear authors,
Thank you for resubmission. The reviewers agree that the results and presentation of the research are interesting but I would like to ask you to very carefully address the issues raised by the referees and revise your paper accordingly.

Reviewer 2 ·

Basic reporting

Thanks for the authors for updating the paper and uploading the datasets. The revised version and rebuttal letter read well to me and clarify most of my questions . I only have two comments. Please see them below.

Experimental design

The deep learning models are used to evaluate the accuracy of review spam detection model. Why do you use deep learning models for evaluation? In my opinion, you should use the ground truth --- the spammer scores described in Section 4.1.3 for evaluation. Is it because you have limited ground truth data? Then you will need to rely on an automatic metric to approximate the ground truth. How much ground truth data do you have? You use the ground truth to train your deep learning models right? What are the performance of the deep learning models? If the deep learning model performs not well, it is not a good approximation of ground truth. If the deep learning model works well, why don't you just use them for spammer detection, as you mentioned they have advantages of using non-handcrafted features.

Validity of the findings

The size for the Daraz dataset should be 3,923. Why did you select only one single product for the review diversification task in Section in Section 5.2? Your proposed method would be better justified and generalized by using reviews from more products.

Additional comments

The authors have provided arguments about using SGD for spammer detection and deep learning models for evaluation answering #4 question in my previous review. However, it is unclear how both methods perform against the ground truth data. Please see my comments at the "experimental design" section. For the review diversification experiments in Section 5.2, it would be good if the authors could provide results on more products but not just one project. I hope the authors could address these two questions in the next version of the paper.

---

## Round 0.3 · accepted · Accept

Dear Dr. Mirza,

I am pleased to inform you that your paper has been accepted for publication in PeerJ Computer Science Journal.

Thank you for submitting your work.

With kind regards,